# Use of Enriched Mine Water to Grow the Cyanobacterium *Arthrospira platensis* in Photobioreactors

**DOI:** 10.3390/foods14101665

**Published:** 2025-05-08

**Authors:** Massimo Milia, Valeria Andreotti, Angelica Giglioli, Viviana Pasquini, Pierantonio Addis, Alberto Angioni

**Affiliations:** Department of Life and Environmental Science, University of Cagliari, 09100 Cagliari, Italy; valeria.andreotti@unica.it (V.A.); giglioli.angelica@gmail.com (A.G.); addisp@unica.it (P.A.); aangioni@unica.it (A.A.)

**Keywords:** *Arthrospira platensis*, mine water, Zarrouk medium, biochemical composition

## Abstract

The demand for sustainable and high-nutritional food sources is forcing the industrial sector to find alternatives to animal proteins. Microalgae and macroalgae showed remarkable protein and bioactive compound content, offering a promising solution for the food industry. However, the high production cost represents the main concern related to microalgae development. Thus, strategies that can reduce production costs, preserve the environment, and improve the nutritional characteristics of microalgae are required. Exploiting water from dismissed mines could lead to energy savings in production by opening new industrial opportunities to produce microalgae. *Arthrospira platensis* (Spirulina) can be grown in open ponds and photobioreactors; the composition of the growth medium and the light radiation could affect its biochemical composition. This work investigated the influence of mine water with the addition of Zarrouk growth medium on the biochemical composition of the final dried Spirulina. The trials were performed in vertical tubular photobioreactors (PBRs) exposed to the same light radiance. Samples were compared with standard growing conditions using distilled water with the addition of Zarrouk medium. Spirulina strains showed good tolerance to medium/high concentrations of Cl^−^, SO_4_^2−^ and nitrogen in mine water. The experiment lasted 12 days, showing significant differences in protein, lipids, and carbohydrates between trials. Spirulina grown in mine water showed higher protein levels, 52.64 ± 2.51 g·100 g^−1^ dry weight. On the other hand, Spirulina grown in distilled water had higher lipids and carbohydrate levels, accounting for 9.22 ± 1.01 and 31.72 ± 1.57 g·100 g^−1^ dry weight. At the end of the experiment, both trials showed similar growth and pigment concentration. The availability of a high amount of mine water at no cost and at the ideal temperature for Spirulina cultivation increases environmental sustainability and reduces production costs. The results in terms of biomass were comparable to those of standard cultivation, whereas proteins showed higher values. Moreover, coupling renewable energy sources can further reduce production costs, with promising industrial and market developments.

## 1. Introduction

Macro- and microalgae can contribute to meeting the next generation’s food demand and environmental sustainability [1]. Despite the increasing market size, microalgal production is still limited compared to the global algal output. The main reason is the high production cost and low yields when scaling up from laboratory to industrial cultivation. Microalgae have remarkable nutritional composition with high-value proteins, lipids, carbohydrates, active enzymes, pigments, sterols, and vitamins [2]. Microalgae and cyanobacteria are currently employed in several applications, such as food, pharmaceutics, agriculture, wastewater treatment, aquaculture, cosmetics, and biofuels [3,4,5,6,7]. In addition, the World Health Organization has labelled microalgae as a superfood, a rich source of biologically active compounds that could be used as functional ingredients [8]. Nowadays, the cyanobacterium *Arthrospira platensis* Gomont, 1892 (Spirulina), due to its characteristics and growth conditions, is the most commercialized species in the bio-resource market, with a 2025 global market of USD 522.6 million and forecast to reach USD 884.6 million by 2032 [9]. Spirulina is considered a superfood due to its macronutrient composition. Proteins range from 55 to 70%, providing all essential aminoacids, whereas lipids and carbohydrates range from 5 to 6%, and 15 to 25%, respectively. It contains chlorophyll a (Chl a), carotenoids, phycobiliproteins, essential fatty acids, polyphenols, and sterols, showing a complete composition for nutritional purposes [10].

Spirulina naturally thrives in alkaline waters of lakes and confined ponds, preferably in warm areas [11]. Domestication can use different cultivation methodologies, including photobioreactors or open ponds, operating in batch or semi-continuous modes. Open ponds in outdoor cultivation conditions show higher duplication times and lower biomass productivity. However, the high variability of these systems (such as temperature and light intensities) can influence the growth performance of cyanobacteria [12]. In semi-continuous production, the cyanobacteria are harvested at the logarithmic phase, producing higher macronutrients. The cultivating system is restored after harvesting to support new growth [10].

Spirulina does not compete with traditional agricultural production; however, various bottlenecks are identified in the cultivation of Spirulina such as the high costs associated with the Zarrouk medium, as well as the alternation of production associated with the variation of environmental conditions in open ponds and extensive tube pipe systems which influence biomass and bioactive compound yields. Furthermore, both methods require the use of large quantities of water, which must be subsequently sanitized before being released into the environment, increasing manpower and energy expenditure and decreasing the market demand. The Zarrouk medium is one of the earliest and most widely used for Spirulina cultivation due to its ability to support high biomass productivity and maintain culture stability; however, numerous studies have evaluated substituting the Zarrouk medium with less expensive substrates capable of producing Spirulina biomass of comparable quality, such as fish farming, piggery, brewery, winery, industrial, and dairy wastewater [13,14,15,16,17,18,19]. Indeed, no final results have been achieved. Laboratory-scale production systems are usually set up with artificial production of light. However, these systems do not consider the impact of energy cost on the overall production and do not make a scalability study of the reactor for industrialization. No studies have been conducted on alternatives to water and energy supplementation for costs and environmental impact reduction.

Mining sites offer extensive areas and water resources that can be converted into aquaculture facilities for cyanobacteria cultivation [20]. Redeveloping decommissioned mining areas into cyanobacteria cultivation sites can represent a significant opportunity to achieve environmental, social, and economic benefits [21]. Water resources have two advantages: they have no cost and a constant temperature, requiring no energy to warm them to the ideal temperature for Spirulina growth, therefore lowering production cost.

The historic coal mine of Monte Sinni (Carbonia-Iglesias, southwestern Sardinia) includes more than 30 km of tunnels and sits above a warm water basin. Daily, about 3000 m^3^ of water is pumped from −500 m underground to avoid flooding the galleries. The water temperature is constant at about 42 °C, reaching 38 °C at the first surface storage tank. Nowadays, its primary destination is the sustenance of a nearby river used to irrigate the cultivated fields in the area [22].

The aim of this study was to verify the feasibility of growing *A. platensis* with the water mine effluent alone or enriched with conventional Zarrouk medium in vertical tubular photobioreactors (PBRs). The trials were set to assess whether the nutrient composition of the mine water was sufficient for Spirulina growth or required enrichment. Moreover, the trials evaluated if the mine water enriched with Zarrouk medium was comparable with standard growing systems. To assess the performance of the cyanobacterium, biomass production, photosynthetic pigment amount, biochemical composition, total lipids, proteins, carbohydrates, neutral detergent fiber (NDF), and ash were evaluated.

## 2. Materials and Methods

### 2.1. Cyanobacteria Culture and Photobioreactors

Livegreen SRL (Arborea, Sardinia, Italy) provided the cyanobacteria strains of *A. platensis*. Pre-culture inocula cultivated in Zarrouk medium [23] were permanently kept in Erlenmeyer flasks in Pyrex glass with a total capacity of 5 L at 30 °C, irradiated with 80 µmol photons·m^−2^·s^−1^, and mixed with continuous aeration.

The experiments were carried out using four vertical PBRs in polymethyl methacrylate with a capacity of 25 L. PBRs were placed in a vertical steel structure in a monitored room. They consisted of an upper cap with three accessways for the filling system, ventilation, and heating and a lower cap with a tap for recovery and cleaning. The filling medium was stored in a 150 L tank connected to the PBRs by a pump system managed with a control panel.

The PBRs were illuminated by three types of LED light lamps (24 h a day) anchored to the support structure and oriented to allow maximum light exposure. The following light spectra characterized LED lamps: (1) white 380–760 nm, maximum 437 nm, and 630 nm; (2) red 625–675 nm, maximum 650 nm; and (3) blue 400–475 nm, maximum 450 nm. All lamps operated simultaneously with the same light intensity of 80 µmol·m^−2^·s^−1^. The light spectrum was selected according to previous finding on photobioreactor experiments [10]. Immersed heating probes maintained the temperature at 33 ± 2 °C (Aquarium Heater, 3613010, Eheim, Thermocontrol, Deizisau, Germany), which is within the optimal 30–35 °C range for Spirulina growth, and the temperature was made affordable using a continuous flow of mine water from the first surface storage tank. The medium’s pH was 11.13 ± 2.14 (RSD%).

The mine water was collected at the first surface storage tank and analyzed in the laboratory’s facilities before the experiments. Heavy metal and metalloid composition examination was carried out by ICPOES analysis following EPAMethod 200.7, Rev. 4.4, and EPA Method 3050B for As. COD was determined by UV–Visible spectrophotometry (ISO 15705:2002) [24], hardness and F^−^ were determined by photometry (ISO/TS 15923-2-2017) [25], NO_3_^−^, NO_2_^−^, NH_4_^+^ SO_4_^2−^, Cl^−^, and PO_4_^3−^ by UV–Visible spectrophotometry (ISO 15923-1:2023) [26].

Three trials were carried out simultaneously with different mediums: (a) deionized water (DWZ) (Control), (b) mine water (MWZ), both enriched with Zarrouk medium (Table 1), and (c) mine water (Table 2) (MW).

### 2.2. Experimental Design

Four PBRs were used for each treatment in batch condition, filled with 20 L of medium. Before transferring Spirulina into the PBRs, an aliquot from the Erlenmeyer flasks was collected to determine dry weight. Therefore, Spirulina was placed in each PBR to reach an initial concentration of 200 mg·L^−1^ dry weight. Control sampling was performed daily, collecting 100 mL of culture from each PBR, dividing the sample into three replicate flasks, and maintaining it under agitation before physical analysis. Before sampling, each PBR was refilled with distilled water to compensate for the evaporation.

Trials lasted 12 days when the late logarithmic phase was achieved. The pH was measured daily with a bench-top pH meter (Hanna Instruments, Rome, Italy).

At the end of the experiment, the entire culture was collected, filtered at 0.25 mm, washed two times with deionized water, and lyophilized. Freeze-dried samples were used for biochemical analysis after being finely ground in a mortar. Each study was carried out in triplicate.

### 2.3. Chemicals

Methanol (MeOH), hydrochloric acid (HCl), sodium hydroxide (NaOH), and chloroform (CHCl_3_) were ultra-residue solvents of analytical grade. Sulfuric acid (96%, 0.5 N), sodium hydroxide (32%, 0.5 N and 1 N), phenol, KCl, CaCl_2_, Na_2_CO_3_, CuSO_4_, Na and K tartrate, D-glucose, α-amylase, and methyl red indicator were reagent grade solvents. Chemical salts used for the Zarrouk medium (Table 1) were of analytical grade, purchased from Sigma Aldrich (Merck KGaA, Darmstadt, Germany). Phosphate buffer solution (PBS, 0.01 M) was prepared in laboratory. Double-deionized water with a conductivity of less than 18.2 MΩ was obtained with a Milli-Q system (Millipore, Bedford, MA, USA).

### 2.4. Biochemical Analysis

The analyses were performed according to Milia et al. [10,27]. Dried biomass weight was assessed on 5 mL of suspension. The samples were filtered (glass microfiber filter membranes 45 mm diameter, Whatman GF/F, Maidstone, England), washed twice with deionized water, oven-dried at 105 °C for 3 h, and weighed on an analytical balance. Chl a and total carotenoids were assessed on 3 mL of the homogenized solutions after the mechanical breakup of the gas vacuoles. The pellet obtained after centrifugation was suspended in 3 mL of methanol. The bright green supernatant was analyzed in a UV–Vis spectrometer after separation by centrifugation of the undissolved cell debris. Phycobiliproteins were determined by UV–Vis spectrometer (Cary 50, Varian Inc., Milan, Italy) at 620, 652, and 750 nm on the solution after cell lysis and centrifugation to precipitate undesirable cellular components. Total proteins were obtained with the Kjeldahl method on 0.5 g of sample, total lipids were determined by ponderal method after saponification, and total carbohydrates were determined by the phenol-sulfuric acid method on 20 mg of sample.

### 2.5. Statistical Analysis

Differences in biomass production, protein, carbohydrate, lipid, ash, NDF, phycocyanin (Pc, mg·L^−1^), allophycocyanin (Apc, mg·L^−1^), chlorophyll a (Chl a, mg·L^−1^), and total carotenoids (Carot, mg·L^−1^) were assessed with permutational analysis of variance (PERMANOVA) tests using Trials (2 fixed levels) and Days (12 fixed levels) as orthogonal sources of variance. PERMANOVA was conducted on Euclidean distance-based resemblance matrices of normalized data, using 999 random permutations of the appropriate units in univariate and multivariate contexts. When significant differences were observed, pairwise tests were also carried out to ascertain patterns of differences among treatments and/or sampling times.

All statistical analyses were performed using the routines included in PRIMER 7+ software 7.0.24 [28]. Multivariate differences were assessed in all photosynthetic pigments (PC, Apc, Chl a, Carot) between trials, with a biplot after a canonical analysis of the principal coordinates (CAPs), which allows identifying an axis through the multivariate cloud of points that is best at separating a priori groups [29].

## 3. Results and Discussion

Spirulina biomass production and its bioproducts are estimated to exhibit significant growth over the next 5–10 years. However, Spirulina cultivation still faces many limitations. The excessive cost of the growth medium, cultivation system, and the extraction technology applied represent the primary limits.

Spirulina cultivation is water- and energy-consuming, affecting both overall expenses and environmental impact.

Despite the capacity of Spirulina to adapt to different environmental conditions by changing its metabolism, selecting the appropriate growth medium for industrial cultivation in raceway open ponds or pipes is mandatory to obtain good yields and the appropriate composition of bioactive compounds.

The Zarrouk medium is one of the earliest and still remains the most widely used for Spirulina cultivation in industrial applications [30]. The high costs of this medium have encouraged researchers to find alternative nutrient sources, together with selected environmental conditions [31,32]. AlFadhly et al. identified critical bottlenecks of Spirulina cultivation, emphasizing temperature, light intensity, and the physicochemical properties of the culture medium as key factors influencing growth performance [32]. Multi-trophic aquaculture systems represent an interesting development according to the Circular Economy Action Plan of the EU. However, the main industrial application is represented by agricultural wastewater bioremediation programs and anthropogenic activity-related water contamination [6,7,33,34,35,36,37]. Moreover, wastewater has two main problems, turbidity and susceptibility to contamination. Turbidity decreases light penetration, influencing microalgae growth and bioactive compounds synthesis. This necessitates effluent dilution, resulting in additional water consumption [38]. Spirulina can uptake chemical contaminants such as heavy metals from wastewater, concentrating them and becoming potentially toxic for food and feed, cosmetics and pharmaceutical applications [39]. In addition, biocontamination of the medium can inhibit the growth of Spirulina, requiring decontamination treatments or environmental adjustments such as high salinity levels and high alkalinity conditions [40,41].

Several studies dealing with the substitution of the growth medium can be found in the literature. Pereira et al. studied the mixotrophic cultivation of Spirulina, replacing part of the standard medium with cheese whey. The authors found increased phenolic compounds with 10% whey and increased growth rates [42]. These results were confirmed by Athanasiadou et al., who reported the need for whey pretreatment by filtration at 1 μm to decrease turbidity and avoid pathogen contamination [19].

Papadopulos et al. investigated the use of brewery wastewater to cultivate Spirulina. The microalga showed good yield and biochemical composition. However, many heterotrophic bacteria were detected, limiting the biomass use in food and feed products [16].

Spennati et al. reported data from the use of winery wastewater; the yield of protein, carbohydrates, and polyphenols was lower than conventional growing conditions, even if the total biomass was similar [17].

Scaling up Spirulina cultures to the volumes required for commercial production with an alternative medium to Zarrouk is not affordable. In this context, the efficiency of cultivation using disused natural resources was evaluated to reduce processing costs and environmental impact.

Dismissed mine sites represent high social and environmental cost. The redevelopment of these sites into an environmentally sustainable and productive microalgae farm represents one of the strategies of the European Commission. The Monte Sinni mine can afford high quantities of water at an optimal temperature for Spirulina cultivation, avoiding heating systems and substantially reducing cultivation cost.

This study explored the use of mine resources as a growth medium for Spirulina in controlled systems. The trials were conducted in vertical PBRs using laboratory equipment; the temperature was kept constant at 33 °C to reproduce the mining site conditions.

Three experiments were set up: mine water (MW), distilled water enriched with Zarrouk medium (DWZ), and mine water enriched with Zarrouk medium (MWZ).

After 24 h from the inoculum, Spirulina in MW separates from the culture medium, forming a gelatinous mass represented by dead cells, and moves to the medium surface. Therefore, it was no longer considered throughout the study. Nitrogen limitation influences microalgae growth and cell density [43]. Moreover, very low nutrient levels in the early exponential phase reduce microalgae’s ability to respond to environmental stresses such as the presence of metals. The mine water (Table 2) is low in nitrogen and has detectable levels of metals. These metals could adversely affect Spirulina growth, stimulating the production of exopolysaccharides to form a protective sheath, incorporating water. Instead of settling to the bottom of the photobioreactor, the dead cell formed a kind of mucilaginous structure on the surface.

During the 12 days of the experiments, the growth (expressed as dry weight) showed distinct patterns in DWZ and MWZ. In DWZ, Spirulina had an almost constant value in the first 5 days and then increased steadily until the 12th day. On the other hand, MWZ showed a wave-like trend till day 5, followed by a consistent upward pattern till day 12. At the end of the experiment, statistical analysis showed substantially equivalent biomass production in the two trials, except on days 1, 3, and 9, which highlighted higher growth in DWZ (Figure 1, Table 3).

The different growth patterns on other days did not affect the overall development of the culture, leading to similar values at the late-logarithmic phase in both trials.

The composition of photosynthetic pigments in the biomass obtained in the experiment (mg·100 mg^−1^ dry weight) showed significant differences between trials only during the first 8 days of the experiment. The values in DWZ were 1.2 times higher for carotenoids, Chl a, and Pc and 1.3 times higher for Apc. At the late-logarithmic phase (plateau), MWZ showed no significant disparities in pigment concentration compared to DWZ, suggesting that the culture adequately adapted to mine water composition.

DWZ showed a constant increase in Chl a and carotenoids with an r^2^ of 0.994 and 0.988, respectively. MWZ showed an irregular increase with an r^2^ of 0.971 and 0.973, respectively. Pc and Apc had an opposite trend with r^2^ of 0.833, and 0.919 and 0.944 and 0.857 in DWZ and MWZ, respectively (Figure 2 and Figure 3, Table 3). Metal cations bind to phycobiliproteins, quenching their light absorbance, thus resulting in lower apparent concentration. The absorbance spectra for APC and PC shows little shift in λmax but some decrease in total absorbance, suggesting that only moderate structural perturbations occur upon metal binding. The mechanism of heavy metal binding to phycobiliproteins is not fully understood, limiting their widespread quantitative applications [44].

MWZ data on macronutrients are consistent with literature values ranging from 55 to 70% protein, 5 to 6% lipid, and 15 to 25% carbohydrate, whereas DWZ had higher lipid and carbohydrate amounts (Table 4 and Table 5) [45,46,47]. NDF and ash content did not differ significantly between trials. Protein-to-carbohydrate ratios increased from 1.45 to 2.05, and protein-to-lipid ratios rose from 5.00 to 8.48 in MWZ, suggesting that the mine water medium, which has a consistent amount of SO_4_^2−^, improved protein synthesis at the expense of carbohydrate and lipid accumulation. This behavior agrees with literature data reporting that supplementation with sulphate salts such as MgSO_4_ and K_2_SO_4_ has been associated with increased biomass yield, protein content, and cysteine and methionine levels, without affecting pigment or lipid profiles [48].

The presence of toxic compounds (e.g., copper) and the unbalanced concentration of inorganic N or P could interfere with the micro-algal-bacterial consortium’s physiology, growth, and metabolism [49]. Therefore, the transition from laboratory to industrial development remains unfinished.

Elevated chloride (Cl^−^) concentrations in the mine water substrate could negatively impact Spirulina growth, decreasing pigments and dry weight in the exponential growth phase of MWZ. Grettenberger et al. reviewed the factors limiting Spirulina growth. These authors highlighted the accumulation of ROS at elevated levels of Cl^−^, negatively affecting photosynthesis [50]. However, in contrast to the higher plants, the cyanobacteria exhibited high capacity to adapt to different environmental conditions and a significantly higher salt stress resistance. This adaptability has been supported in many studies conducted on natural populations of Spirulina [51,52] in a laboratory environment [53,54,55] and outdoor mass cultures [51,54] exposed to different growth conditions. This fact can support the equal performance of Spirulina in DWZ and MWZ after the 12-day experiments.

The higher levels of lipids and carbohydrates in DWZ could be explained considering the lower levels of total nitrogen with respect to the MWZ medium. Nitrogen limitation increased during the starvation phase and stress conditions, leading to a surplus of carbon derived compounds from photosynthesis and glycolysis is converted in lipid and carbohydrate accumulation [48,56]. This behavior was highlighted in wastewater from dairy, brewery, and winery industries where the low nitrogen concentration of the different mediums led to a decrease in the protein content and an increase in lipids and carbohydrates [16,17,39].

## 4. Conclusions

The mine water represents an inexhaustible water source at a constant temperature, ideal for growing Spirulina with significant energy savings and reductions in production costs. The mineral profile of mine water, which is characterized by a high concentration of SO_4_^2−^, can potentially enhance Spirulina’s protein content and overall nutritional profile. The pigment concentration analysis showed similar values at the end of the experiment, indicating that the mine water could be successfully used for Spirulina cultivation. Spirulina strains confirmed the high adaptability to different environmental conditions during growth, allowing comparable final dry weight yields to the standard growing conditions represented by distilled water with the addition of the Zarrouk medium.

However, when dealing with water from mining sites, it is imperative to conduct rigorous and continuous water quality monitoring to manage risks from potential contaminants, such as trace metals or excess ions, to ensure that these factors do not compromise the safety and nutritional integrity of the biomass.

Moreover, integrating these techniques with renewable energy production systems can further save production costs and marketing opportunities.

The Spirulina biomass cultivated in mine water could have various applications. Its protein-rich content makes it suitable for nutritional applications, including use in animal feed and aquaculture, as well as a phagostimulant compound, while its bioactive compounds, such as phycocyanin and antioxidants, could be valuable for pharmaceutical, nutraceutical, and cosmetic applications. From an economic perspective, this strategy offers a competitive advantage by reducing dependency on conventional water sources and supporting the large-scale production of Spirulina-based products. From an environmental perspective, utilizing a readily available resource mitigates water wastage and promotes resource efficiency in accordance with European Blue Economy and Green Deal principles. From a social perspective, this approach could drive job creation and stimulate local economies in mining regions.

## Figures and Tables

**Figure 1 foods-14-01665-f001:**
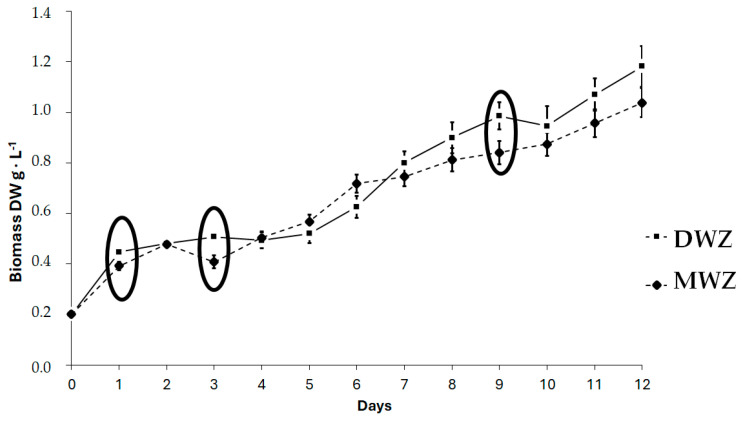
Biomass evolution (g·L^−1^) during Spirulina growth in two trials. DWZ used bi-distilled water with Zarrouk medium, and MWZ used mine waters with Zarrouk medium. Error bars indicate the standard deviation. Points inside the circles show statistical differences.

**Figure 2 foods-14-01665-f002:**
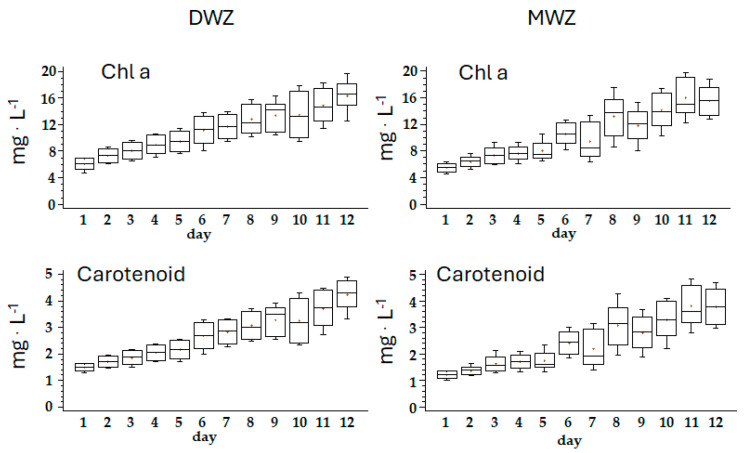
Box-and-whisker plot of the variation of Chl a and carotenoids (mg·L^−1^) in the two trials along the 12 days of the experiment. Error bars indicate standard deviation.

**Figure 3 foods-14-01665-f003:**
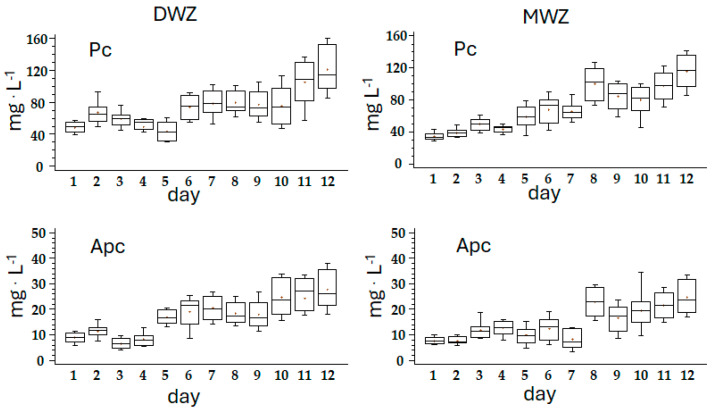
Box-and-whisker plot of the variation of PC and APC (mg·L^−1^) in the two trials along the 12 days of the experiment. Error bars indicate standard deviation.

**Table 1 foods-14-01665-t001:** Chemical composition of the Zarrouk medium.

Ingredients	g·L^−1^
NaHCO_3_	16.80
K_2_HPO_4_	0.50
NaNO_3_	2.50
K_2_SO_4_	1.00
NaCl	1.00
MgSO_4_·7H_2_O	0.20
EDTA-Na_2_·2H_2_O	0.08
CaCl_2_·2H_2_O	0.04
FeSO_4_·2H_2_O	0.01
H_3_BO_3_	2.860
MnCl_2_·4H_2_O	1.810
ZnSO_4_·7H_2_O	0.222
Na_2_MoO_4_·H_2_O	0.007
CuSO_4_·5H_2_O	0.079

**Table 2 foods-14-01665-t002:** Chemical composition of the mine water.

Physicochemical Properties	Values (conc ± RSD%)
COD (mg·O_2_^−1^)	5.80 ± 0.10
NO_3_^−^ (mg·L^−1^)	2.07 ± 0.03
NO_2_^−^ (mg·L^−1^)	0.02 ± 0.01
NH_4_^−^ (mg·L^−1^)	3.34 ± 0.49
SO_4_^2−^ (mg·L^−1^)	547.2 ± 1.31
Hardness °F	19.4 ± 0.5
Cl^−^ (mg·L^−1^)	606.6 ± 1.53
PO_4_^3−^ (mg·L^−1^)	4.72 ± 0.07
Fe (mg·L^−1^)	<0.001
Ba (mg·L^−1^)	0.33 ± 0.01
B (mg·L^−1^)	0.31 ± 0.03
Al (mg·L^−1^)	0.014 ± 0.003
As (mg·L^−1^)	0.035 ± 0.005
Zn (mg·L^−1^)	0.004 ± 0.001
F^−^ (mg·L^−1^)	5.85 ± 0.02
pH	8.3

**Table 3 foods-14-01665-t003:** Results of univariate PERMANOVA testing for the effects of different culture media (DWZ, MWZ) and the days of the experiment (12 days) on dry weight (g·L^−1^) and photosynthetic pigments (Pc, Apc, Chl a, Carot) (mg L^−1^).

		Dry Weight		Chl *a*		Carot		Pc		Apc	
	Days	t	P (MC)	t	P (MC)	t	P (MC)	t	P (MC)	t	P (MC)
DWZ vs. MWZ	1	2.460	*	1.854	ns	4.638	***	6.278	***	1.651	ns
	2	0.124	ns	2.571	*	4.341	***	6.354	***	4.729	***
	3	2.986	**	1.410	ns	1.964	*	2.644	*	4.770	***
	4	0.239	ns	2.561	*	3.014	*	1.143	ns	3.988	***
	5	1.027	ns	2.351	*	2.819	**	2.947	**	5.553	***
	6	1.619	ns	0.646	ns	1.238	ns	0.946	ns	3.134	**
	7	0.919	ns	2.311	*	2.663	*	2.294	*	6.723	***
	8	1.140	ns	0.401	ns	0.042	ns	2.757	*	2.054	ns
	9	2.038	*	1.617	ns	1.941	ns	1.074	ns	0.656	ns
	10	0.786	ns	0.482	ns	0.150	ns	0.538	ns	1.831	ns
	11	1.347	ns	1.062	ns	0.317	ns	0.763	ns	0.709	ns
	12	1.462	ns	0.887	ns	1.776	ns	0.508	ns	0.993	ns

* *p* ≤ 0.05, ** *p* ≤ 0.01, *** *p* ≤ 0.001, ns = non-statistically significant.

**Table 4 foods-14-01665-t004:** Biochemical composition (g·100 g^−1^ ± RSD%) of Spirulina in DWZ and MWZ.

	Lipids	Carbohydrates	Proteins	NDF	Ashes
DWZ	9.22 ± 1.01	31.72 ± 1.57	46.03 ± 2.14	2.69 ± 0.37	9.00 ± 0.32
MWZ	6.25 ± 0.88	25.66 ± 2.12	52.64 ± 2.51	2.34 ± 0.11	8.18 ± 1.41

**Table 5 foods-14-01665-t005:** Results of univariate PERMANOVA testing for the effects of different culture media (DWZ, MWZ) on the lipids, carbohydrates, proteins, NDF, and ashes (g·100 g^−1^).

	Source	DF	MS	Pseudo-F	P (MC)
Lipids	Trial (Tr)	1	5.36	19.6	**
	Res	6	0.27		
Carbohydrates	Trial (Tr)	1	5.45	21.2	**
	Res	6	0.26		
Proteins	Trial (Tr)	1	5.09	16.02	**
	Res	6	0.32		
NDF	Trial (Tr)	1	2.48	3.29	ns
	Res	6	0.75		
Ashes	Trial (Tr)	1	1.24	1.3	ns
	Res	6	0.95		

DF = degree of freedom; MS = mean squares; Pseudo-F = F; P (MC) = probability level after Monte Carlo simulations. ** = *p* < 0.01; ns = *p* > 0.05.

## Data Availability

The original contributions presented in the study are included in the article. Further inquiries can be directed to the corresponding authors.

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
