# Peer review of "Use of Enriched Mine Water to Grow the Cyanobacterium *Arthrospira platensis* in Photobioreactors"

_foods, 2025, doi:10.3390/foods14101665_

Round 1
Reviewer 1 Report
Comments and Suggestions for Authors
Dear Authors,
I have reviewed your manuscript entitled “Impact of growth medium on the biochemical composition of the cyanobacterium Arthrospira platensis grown in photobioreactors.” I found the use of coal mining effluent particularly interesting; however, this aspect needs to be better explored in the Introduction to support the novelty and originality of your work with this specific strain.
Moreover, the Results and Discussion section requires substantial improvement. Currently, it lacks a coherent and well explained mechanistic discussion that would build a complete narrative around your findings. The discussion is too brief and fails to contribute meaningfully to the state of the art. As it stands, it raises more questions than it answers for the reader.
Below are specific comments to help improve the manuscript:
Keywords:
Avoid repeating words already present in the title.
Title:
Consider referencing the use of sewage water from decommissioned mines to make the title more specific. The current title is too general.
Abstract:
At the end of the abstract, strengthen the take-home message. What is the key practical application or most important contribution of your study that readers should remember?
Line 48:
Correct the citation format.
Lines 52–54:
Avoid very short (e.g., three line) paragraphs. Reorganize for clarity and flow.
Introduction:
Discuss current bottlenecks in Spirulina cultivation, both in laboratory and realworld scenarios.
Link your approach using mining effluent to these bottlenecks.
Clarify the novelty and originality of your work.
End the Introduction with a clear hypothesis regarding the use of mining wastewater for cyanobacterial cultivation:
What nutrients are expected to be present?
How might these nutrients support cyanobacterial growth?
Briefly outline what was done to test this hypothesis.
Replace the term “microalgae” with “cyanobacteria” for taxonomic precision.
Lines 107–108:
Avoid twoline paragraphs; merge or expand with context.
Coal mining effluent composition:
Provide details on the source and characterization of the effluent.
Was it analyzed by the authors?
Or were data taken from previous studies?
Lines 155–157:
Clearly state which analytical techniques were used.
It is insufficient to cite methods from other works without explaining what was done.
Results and Discussion:
Combine these sections to improve conciseness and avoid redundancy.
Lines 175–176:
Clarify the meaning of “collapse.”
If nutrient limitation occurred, biomass should plateau rather than decline.
If cell lysis happened within 24 hours, this may indicate another factor beyond nutrient limitation.
This point will certainly raise questions from readers, so it must be clearly addressed.
Also, include the data on biomass decline in the relevant figure(s).
Figure 1:
Correct the Y axis formatting: use dots instead of commas (e.g., “1.0” not “1,0”) to denote decimals.
Add a trend line or curve for Trial 2 to match the other trials.
Discussion Section: Replace “Trial 1,” “Trial 2,” and “Trial 3” with descriptors reflecting each group's actual treatment or condition.
Strengthen the mechanistic explanation of your results:
Link your findings with theories and previous studies.
Formulate a coherent and wellsupported narrative that explains the observed trends.
Lines 320–324:
Elaborate on how SO₄²⁻ enhances protein synthesis in cyanobacteria, particularly Spirulina.
Provide a biochemical or physiological explanation supported by literature.
Lines 325–327:
Similarly, expand the discussion to explain how the different group compositions influenced the biochemical profile of the biomass.
Lines 332–333:
Avoid oversimplification. This explanation is too vague and will likely raise questions among readers. Offer a more in depth, specific interpretation.
Ensure that a full mechanistic discussion of the findings and their industrial significance is provided.
Author Response
R1 Keywords:
Avoid repeating words already present in the title.
A1 Keywords should help people search for an article to find easily the main issues, therefore we think that these could be the most useful:
Arthrospira platensis; mine water; Zarrouk medium; biochemical composition. (Line 38)
R2 Title:
Consider referencing the use of sewage water from decommissioned mines to make the title more specific. The current title is too general.
A2 The title was changed: Use of enriched mine water to grow the cyanobacterium Arthrospira platensis in photobioreactors. (Lines 2-4)
R3 Abstract:
At the end of the abstract, strengthen the take-home message. What is the key practical application or most important contribution of your study that readers should remember?
The availability of high amount of mine water at no cost and at the ideal temperature for Spirulina cultivation increases environmental sustainability and reduces production costs. The results in terms of biomass were comparable to standard cultivation whereas proteins showed higher values. Moreover, coupling renewable energy sources can further reduce production costs, with promising developments in the industrial and the market sector. (Lines 31-37)
R4 Line 48:
Correct the citation format.
A4 The citation was corrected. (Line 48)
R5 Lines 52–54:
Avoid very short (e.g., three line) paragraphs. Reorganize for clarity and flow.
A5 The text was modified, as requested. (Lines 55-59)
R6 Introduction:
Discuss current bottlenecks in Spirulina cultivation, both in laboratory and real world scenarios. Link your approach using mining effluent to these bottlenecks. Clarify the novelty and originality of your work.
End the Introduction with a clear hypothesis regarding the use of mining wastewater for cyanobacterial cultivation: What nutrients are expected to be present? How might these nutrients support cyanobacterial growth? Briefly outline what was done to test this hypothesis. Replace the term “microalgae” with “cyanobacteria” for taxonomic precision.
A6 The text was implemented, as requested. (Lines 70-89)
R7 Lines 107–108:
Avoid two-line paragraphs; merge or expand with context.
A7 The text was modified, as requested. (Lines 136-139)
R8 Coal mining effluent composition:
Provide details on the source and characterization of the effluent. Was it analyzed by the authors? Or were data taken from previous studies?
A8 The text was implemented, as requested. (Lines 140-146)
R9 Lines 155–157:
Clearly state which analytical techniques were used.
It is insufficient to cite methods from other works without explaining what was done.
A9 Dear reviewer, some reviewers ask to delete all the information that repeats previous studies, others request this information. I agree with you that it is better to put the minimum information to explain what was really done, therefore we implemented the text as requested. (Lines 209-222)
R10 Results and Discussion:
Combine these sections to improve conciseness and avoid redundancy.
A10 The sections were merged, and the text was modified as requested.
R11 Lines 175–176:
Clarify the meaning of “collapse.” If nutrient limitation occurred, biomass should plateau rather than decline. If cell lysis happens within 24 hours, this may indicate another factor beyond nutrient limitation. This point will certainly raise questions from readers, so it must be clearly addressed. Also, include the data on biomass decline in the relevant figure(s).
A11 the text was modified to better express the behaviour of Spirulina in trial 3 (MW). (Line 240-248)
R12 Figure 1:
Correct the Y axis formatting: use dots instead of commas (e.g., “1.0” not “1,0”) to denote decimals. Add a trend line or curve for Trial 2 to match the other trials.
A12 The figure was modified, as requested. (Line 260)
R13 Discussion Section: Replace “Trial 1,” “Trial 2,” and “Trial 3” with descriptors reflecting each group's actual treatment or condition.
A13 The text was modified, as requested.
R14 Strengthen the mechanistic explanation of your results:
Link your findings with theories and previous studies. Formulate a coherent and well supported narrative that explains the observed trends.
A14 The text was modified, as requested.
R15 Lines 320–324:
Elaborate on how SO₄²⁻ enhances protein synthesis in cyanobacteria, particularly Spirulina. Provide a biochemical or physiological explanation supported by literature.
A15 The text was modified, as requested.
R16 Lines 325–327:
Similarly, expand the discussion to explain how the different group compositions influenced the biochemical profile of the biomass.
A16 The text was modified, as requested.
R17 Lines 332–333:
Avoid oversimplification. This explanation is too vague and will likely raise questions among readers. Offer a more in-depth, specific interpretation.
A17 The text was modified, as requested.
R18 Ensure that a full mechanistic discussion of the findings and their industrial significance is provided.
A18 The text was modified, as requested.
Reviewer 2 Report
Comments and Suggestions for Authors
Review questions for the manuscript following
1.What is the scientific name of the cyanobacterium studied in this research?
2.Why is Arthrospira platensis (Spirulina) considered a promising alternative protein source?
3.What are the primary limitations of large-scale microalgae production?
4.How does this study propose to reduce the production cost of microalgae cultivation?
5.What unconventional water source was used in this study for Spirulina cultivation?
6.What is the Zarrouk growth medium, and why was it used?
7.What was the control growth medium in this study?
8.What type of cultivation system was used for the experiments?
9.How long did the experimental trials last?
10.What three main biochemical components were compared between the two Spirulina cultivation conditions?
11.Which growth medium resulted in a higher protein content in Spirulina?
12.What were the protein levels (in g·100g⁻¹ DW) in Spirulina grown with coal mine water?
13.Which Spirulina sample showed higher levels of lipids and carbohydrates?
14.What were the lipid and carbohydrate concentrations in Spirulina grown in distilled water?
15.What ions present in coal mine water did Spirulina show good tolerance to?
16.Did both trials show similar growth and pigment concentrations at the end of the experiment?
17.How can the use of coal mine water in Spirulina cultivation benefit the environment?
18.What are some industrial or economic benefits of using coal mine water and renewable energy in Spirulina cultivation?
19.How does light radiation influence the biochemical composition of Arthrospira platensis?
20.Based on this study, how might Spirulina cultivation be optimized for industrial food applications?
Author Response
R1: What is the scientific name of the cyanobacterium studied in this research?
A1: The scientific name is Arthrospira platensis, Line 51
R2: Why is Arthrospira platensis (Spirulina) considered a promising alternative protein source?
A2: The text was implemented. (Lines 55-60)
R3: What are the primary limitations of large-scale microalgae production?
A3: The text was implemented. (Lines 70-89)
R4: How does this study propose to reduce the production cost of microalgae cultivation?
A4: The text was implemented. (Lines 93-96)
R5: What unconventional water source was used in this study for Spirulina cultivation?
A5: Generally, in aquaculture systems it is used distilled or purified water. Only a few studies reported the use of mine water or mine wastewater to cultivate microalgae (References 20-21)
R6: What is the Zarrouk growth medium, and why was it used?
A6: The text was implemented. (Lines 79-81)
R7: What was the control growth medium in this study?
A7: The trials with mine water (MW, and MWZ) were compared with the control trial in deionized water enriched with Zarrouk DWZ. (Lines 148-150)
R8: What type of cultivation system was used for the experiments?
A8: The cultivation system is reported in the paragraph experimental design. (Lines 184-197)
R9: How long did the experimental trials last?
A9: In the paragraph experiment design, it is reported the trials endure. (Line 192)
R10: What three main biochemical components were compared between the two Spirulina cultivation conditions?
A10: All biochemical components were compared to evaluate the influence of the different cultivation conditions. Among them only proteins, lipids, and carbohydrates showed at the end of the trials a significative difference as reported in the text.
R11: Which growth medium resulted in a higher protein content in Spirulina?
A11: Coal mine water enriched with Zarrouk (MWZ) resulted in high protein content (Table 4, 5).
R12: What were the protein levels (in g·100g⁻¹ DW) in Spirulina grown with coal mine water?
A12: The protein levels are reported in table 4.
R13: Which Spirulina sample showed higher levels of lipids and carbohydrates?
A13: The lipid and carbohydrate levels are reported in table 4.
R14: What were the lipid and carbohydrate concentrations in Spirulina grown in distilled water?
A14: The lipid and carbohydrate levels are reported in table 4.
R15: What ions present in coal mine water did Spirulina show good tolerance to?
A15: Mine water composition showed the presence of various ions (Table 2). The amounts of single ions are below the reported concentration affecting Spirulina growing performance. The ion sulphate (SO₄-²) has concentration which can improve growth and biochemical status of S. platensis. The text was improved.
R16: Did both trials show similar growth and pigment concentrations at the end of the experiment?
A16: The text was improved. Figure 1, 3, 4, and Table 3.
R17: How can the use of coal mine water in Spirulina cultivation benefit the environment?
A17: Various bottlenecks are identified in the cultivation of Spirulina, among these, water supply is a big issue. Open ponds and extensive tube pipe systems require the use of massive quantities of water for cultivation and for cleaning. The water after the use must be subsequently sanitized before being released into the environment, increasing manpower and energy expenditure, and decreasing the market demand. No studies have been conducted on alternatives to water and energy supplementation for costs and environmental impact reduction.
The exploitation of warm mine water from dismissed mines could lead to energy savings in production, an enhancement in environmental sustainability, and the promotion of resource efficiency in accordance with European Blue Economy and Green Deal principles.
R18.What are some industrial or economic benefits of using coal mine water and renewable energy in Spirulina cultivation?
A18: Please see A17.
R19: How does light radiation influence the biochemical composition of Arthrospira platensis?
A19: Light radiation experiments are not considered in this paper. Many papers in literature can be found dealing with the light influence on biochemical composition. The light spectrum applied in the trials was selected according to previous finding on photobioreactors experiments [10].
R20.Based on this study, how might Spirulina cultivation be optimized for industrial food applications?
A20: Spirulina based products have many applications, such as food and feed, cosmetics, pharmaceuticals, fertilizers. The cultivation system proposed in this paper is not limited to the food industry but could be applied to different sectors.
Open ponds and extensive tube pipe systems involve the use of high water and energy resources. They also have a significant impact on the environment, particularly in terms of wastewater treatment and discharge.
The mine water represents an inexhaustible water source at a constant temperature, also when brought to the surface, ideal for growing Spirulina with significant energy savings and reductions in production costs. Nowadays, mine waters are discharged to nearby canals to avoid flooding of the galleries. Mine water can avoid heating systems and substantially reduces the energy cost.
From an economic perspective, this strategy offers a competitive advantage by reducing dependency on conventional water sources and supporting the large-scale production of Spirulina-based products. From an environmental perspective, utilizing a readily available resource mitigates water wastage and promotes resource efficiency in accordance with European Blue Economy and Green Deal principles. From a social perspective, this approach can potentially drive job creation and stimulate local economies in mining regions.
The phases related to biomass processing are highly energy consuming. Therefore, the coupling of green energy production systems should be a prerequisite for industrial feasibility.
Round 2
Reviewer 1 Report
Comments and Suggestions for Authors
The corrections were made accordingly. Only check minor revisions regarding the correct use of “dry weight” and “dried weight” throughout the manuscript.
Reviewer 2 Report
Comments and Suggestions for Authors
I have no further concerns.